# Using deep neural networks to evaluate object vision tasks in rats

**Kasper Vinken** [1,2]*, **Hans Op de Beeck** [3]

**1** Department of Ophthalmology, Children's Hospital, Harvard Medical School, Boston, Massachusetts, United States of America, **2** Laboratory for Neuro- and Psychophysiology, KU Leuven, Leuven, Belgium, **3** Department of Brain and Cognition & Leuven Brain Institute, KU Leuven, Leuven, Belgium

* kasper.vinken@childrens.harvard.edu

**Data Availability Statement:** The authors confirm that all data underlying the findings are fully available without restriction. The experimental data analyzed in the current manuscript have previously been published elsewhere (https://doi.org/10.1073/pnas.0811583106, https://doi.org/10.1016/j.cub.

## Abstract

In the last two decades rodents have been on the rise as a dominant model for visual neuroscience. This is particularly true for earlier levels of information processing, but a number of studies have suggested that also higher levels of processing such as invariant object recognition occur in rodents. Here we provide a quantitative and comprehensive assessment of this claim by comparing a wide range of rodent behavioral and neural data with convolutional deep neural networks. These networks have been shown to capture hallmark properties of information processing in primates through a succession of convolutional and fully connected layers. We find that performance on rodent object vision tasks can be captured using low to mid-level convolutional layers only, without any convincing evidence for the need of higher layers known to simulate complex object recognition in primates. Our approach also reveals surprising insights on assumptions made before, for example, that the best performing animals would be the ones using the most abstract representations–which we show to likely be incorrect. Our findings suggest a road ahead for further studies aiming at quantifying and establishing the richness of representations underlying information processing in animal models at large.

## Author summary

Despite years of investigating object recognition in rodents, it remains unclear to what extent their visual system supports a capacity for high-level, abstract representations. Here, we used computational deep neural network models to assess which level of abstraction is required to reproduce rodent behavior in several studies, and which level matches representations in higher rodent visual areas. We found that both behavioral and neural data support mid-level representations at best, and show that certain evidence available in the literature may not provide as strong support for invariant visual object recognition as previously thought. Going forward, our findings suggest that computational models could serve as a principled benchmark for evaluating the richness of information processing across species and for designing experiments to push the boundaries of animal models.

2018.02.037, https://doi.org/10.1523/JNEUROSCI.
3663-13.2014, and https://doi.org/10.1093/cercor/
bhw111). All relevant data are either contained
within the original manuscripts or available on the
following OSF repository: http://doi.org/10.17605/
OSF.IO/4W39D.

**Funding:** This work was supported by Research
Foundation Flanders, Belgium, www.fwo.be
(postdoctoral fellowship of K.V.); by KU Leuven
Research Council, www.kuleuven.be (C14/16/031
of H.O.d.B.); by Excellence of Science (EOS), www.
eosprogramme.be (grant HUMVISCAT of H.O.d.
B.). The funders had no role in study design, data
collection and analysis, decision to publish, or
preparation of the manuscript.

**Competing interests:** The authors have declared
that no competing interests exist.

## Introduction

Two decades ago, macaque monkeys were uncontested as the primary animal model for study-ing visual processing at the cortical level. Since then, rodents have become increasingly popu-lar, in particular because developments in neurotechnology such as cell-level imaging and optogenetics capitalized upon genetic rodent models. A major question emerging from this evolution is how far we can take rodents as a model for the more complex aspects of visual information processing.

The most promising evidence comes from rats. Several high-profile studies have docu-mented seemingly complex visual object recognition capabilities in rats, using classification tasks with computer-rendered objects [1–5] or natural videos [6]. In these experiments, rats show the ability to recognize objects despite various transformations in viewing conditions such as position, size, and viewpoint. Neurophysiological recordings have revealed neural responses in lateral extrastriate cortex that might underlie these behavioral abilities [7,8]. To cite [9], "The picture emerging from this survey is very encouraging with regard to the possi-bility of using rats as complementary models to monkeys in the study of higher-level vision."

However, the extent to which previous object recognition experiments in rats probed higher-level vision has never been tested empirically. That is, to what extent did these classifi-cation tasks actually require an abstract, invariant representation of the visual stimuli? From previous work we know that rats are experts at finding and using the lowest-level feature that is predictive of a correct response in a discrimination task [10]. Thus, in order to appreciate the capabilities of the rodent visual system, it is critical to understand the minimum level of abstraction that is required to solve the tasks that these animals are able to perform. Already from the first landmark study by Zoccolan et al. [1], it has been argued that the behavior is unlikely to be based upon representations found in the primary visual cortex (V1). However, the level of invariance found in the later stages of the primate ventral stream may not be neces-sary to explain generalization in rodent object vision experiments, as even similarly small sized Marmoset monkeys far outperformed rats on the same task [11]. In primates there is a multi-step progression of representations with increasing levels of abstraction beyond V1. The first steps still retain properties such as local receptive fields and a retinotopic organization. In the latest stages, invariant representations emerge where objects and categories are easily separable (**Fig 1**A) [12]. If not based on a V1-like representation, then what level of abstraction would support rodent behavior in object vision tasks?

The difficulty in answering this question is that there is no clear cut definition of what con-stitutes different levels of intermediate representations between V1 and a high-level represen-tation specialized in object recognition. Here we demonstrate the value of a computational framework to address this issue. Advances in deep learning have brought about convolutional deep neural networks (DNNs) that allow to simulate this hierarchical information processing. It turns out that DNNs trained on object recognition learn a cascade of representations similar to the ventral stream in monkeys and humans [14–22] and capture important aspects of object perception [15,18,23]. The architecture of these models consists of a series of layers that trans-form pixel inputs into increasingly abstract representations where objects categories are increasingly separable (**Fig 1**B and 1C). It is important to note that the current DNN models do not capture all aspects of object vision in primates (for example see [24,25]), yet they cap-ture ventral stream processing better than any currently available model type [26]. A frame-work based on pre-trained DNNs has been used to model visual object recognition behavior in primates, accurately capturing object- but not image-level error patterns [27].

Here we used the same DNN models as a principled framework to assess the level of abstraction required to explain the data obtained from rats in object recognition tasks,

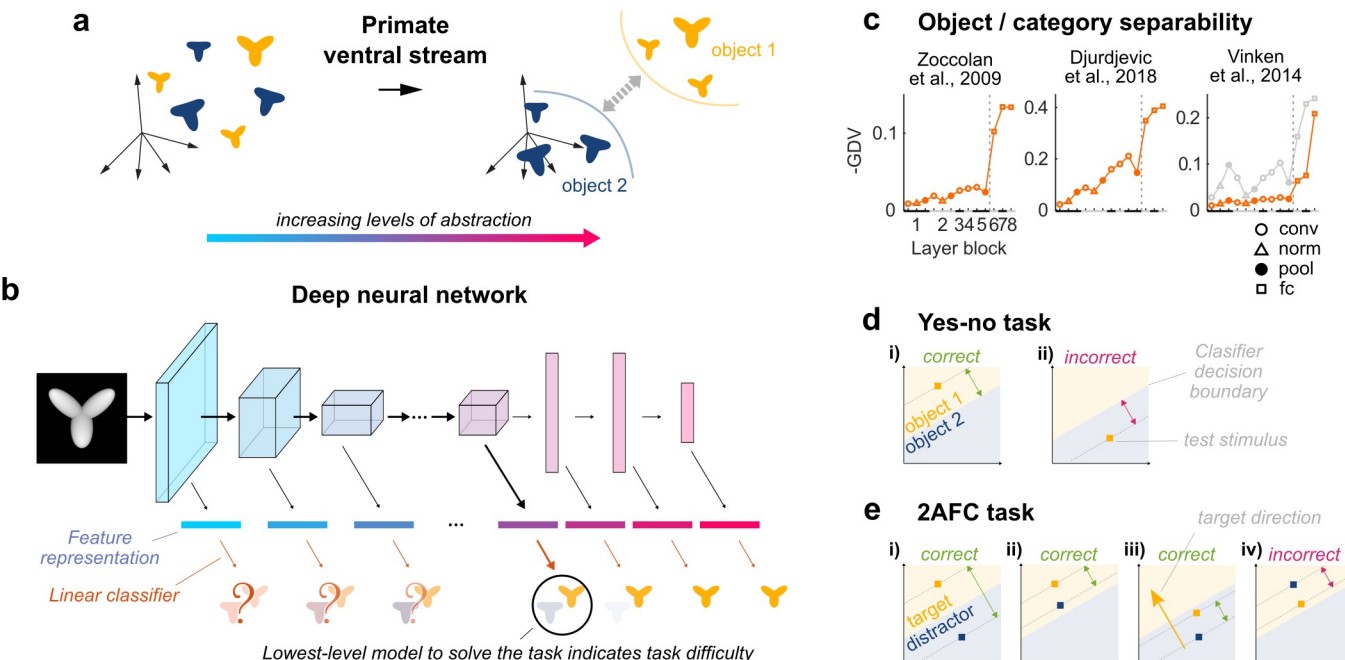

**Fig 1. Evaluating visual object recognition tasks using DNN-based models.** (**a**) The primate ventral stream is thought to transform visual representations where objects are entangled, into more abstract visual representations where objects are increasingly separable [12]. (**b**) A feedforward DNN trained in object categorization of natural images, separated in a sequence of models with increasingly higher-level representations, by training separate fully connected decoder layers (i.e., linear classifiers) on top of each DNN layer's feature representation. (**c**) Higher-level DNN representations (AlexNet) show increased separability of visual stimuli according to object identity [for the stimuli of [1] (left) and of [5] (middle)] and category [for the stimuli of [6] (right; orange: natural distractors; grey: scrambled distractors)]. Separability was quantified as the sign-reversed Generalized Discrimination Value (GDV [13]), which is 0 for non-separable classes and 1 for perfectly separable classes. Black and grey bars on the X-axis indicate layer blocks and markers indicate layer types (see legend insert); the division between convolutional and fully connected layer blocks is indicated by a dashed line. (**d**) For yes-no tasks, correct or incorrect classification of a test stimulus (square marker) was evaluated by comparing the output label of the classifier with the true object label. (**e**) For 2AFC tasks, both the position of the target (yellow marker) and the distractor (blue marker) relative to the decision boundary were taken into account: for a correct response, the target stimulus needed to be positioned more towards the direction of the target side of the decision boundary (see yellow arrow in iii) than the distractor stimulus.

focusing upon several studies suggesting a high level of invariance and abstraction. We evaluated for each DNN layer's representational space whether a linear classifier could generalize from the stimuli used in the training phase of experiments in rats, to the novel stimuli used in the test phase. Thus, rather than asking whether the stimuli of a given experiment are linearly separable (which is likely true for a small amount of stimuli in a very high dimensional space), we are asking whether a linear classifier easily finds a solution for the training set that generalizes well to the test set. That is, we are asking how well each DNN layer's representational space lends itself to the task the rats were probed with. We reasoned that if generalization as observed in rodents is successful in early DNN layer representations, the task does not require high-level, invariant object representations, and thus one should be careful in interpreting generalization performance of rodents as evidence for invariant object recognition. The findings are very consistent across datasets: data from earlier studies can be explained by low to mid-level convolutional representations that fall short of the representations that underlie object recognition in primates. To connect these results back to the rodent visual system, we compared the representational geometry between DNN layers and several visual areas along the putative rodent ventral stream. Consistent with the assessment of behavioral studies, mid-level convolutional layers mapped best onto the higher visual areas in rodents.

## Results

We focus on three studies: the first showed that rats can learn to discriminate two objects invariant to changes in size and azimuth-rotation [1], the second found that rats employ different object recognition strategies that seem to vary in complexity [5], the third showed that rats are capable of ordinate-level categorization of natural videos [6]. We evaluated these experiments by modeling the behavioral tasks using pre-trained DNNs. Computational models based on pre-trained DNNs have been used before to model primate object recognition experiments, by replacing the original decoder of the DNN with a new linear decoder trained on the experimental task [27,28]. The linear decoder layer simulates what a readout neuron/area could decode if it had access to the same visual representation as the penultimate DNN layer. The decoder itself does not add any further non-linear processing and thus requires object representations to already be disentangled in order to be able to successfully generalize in an object recognition task. Such disentangled object representations have been demonstrated in inferotemporal cortex [29]. Linear decoders trained on neural representations in inferotemporal cortex are sufficient to predict core human object recognition performance [30], proving that the framework of a linear decoder trained on object representations is sufficient to capture object recognition behavior.

Because the new decoder in models of primate object recognition received its input from the penultimate DNN layer [27,28], these models used the full depth of the original DNN and had access to a very high-level representation. However, if more of the top layers of the original DNN are removed, the new decoder will be trained on a layer that is lower in the hierarchy of the original DNN, resulting in a different model, which has access to a relatively lower-level representation. Thus, instead of training only one model that uses the full depth of the DNN, we constructed several models, each removing a different number of top layers of the pre-trained DNN before adding the new decoder (**Fig 1**B). The simplest model retained only the first layer of the DNN, a second model retained the first two layers, and so on. As the sequence of layers contains increasingly more abstract representations that are increasingly more useful for invariant visual object recognition, each successive model simulates a visual system which has access to increasingly higher level representations that each map best onto successive stages of the primate ventral stream [15,16,31]. Indeed, for the stimulus sets considered in the present study we found that higher DNN layers showed increasingly better separation for object identity and category (**Fig 1**C), even though these models were trained on ImageNet [32].

Testing and comparing the performance of these models on behavioral experiments of visual object recognition in rats, provides information about the task difficulty in terms of the level of DNN processing steps that would be required to obtain a representational space that lends itself well to the task. For example, if a linear decoder model based on a certain DNN layer's outputs successfully generalizes in a particular task, this suggests that the DNN layer's representation is sufficiently abstract to capture the level of invariance probed by the experiment. Alternatively, a model that fails to generalize suggests that additional non-linear processing may be required to obtain a higher level of invariance and solve the task.

### Any DNN layer can account for size and rotation tolerance in a task with two rendered objects

We started by assessing the first landmark paper that reported evidence for invariant object recognition in rats [1]. In this study, rats were first trained to discriminate between two different objects and then to tolerate variations in size (15 to 40 degrees of visual angle) and azimuth-rotation (60˚ left to 60˚ right), using a yes-no task in an operant box with liquid rewards (see Materials and Methods, Behavior, Task paradigms). At each trial, one object was

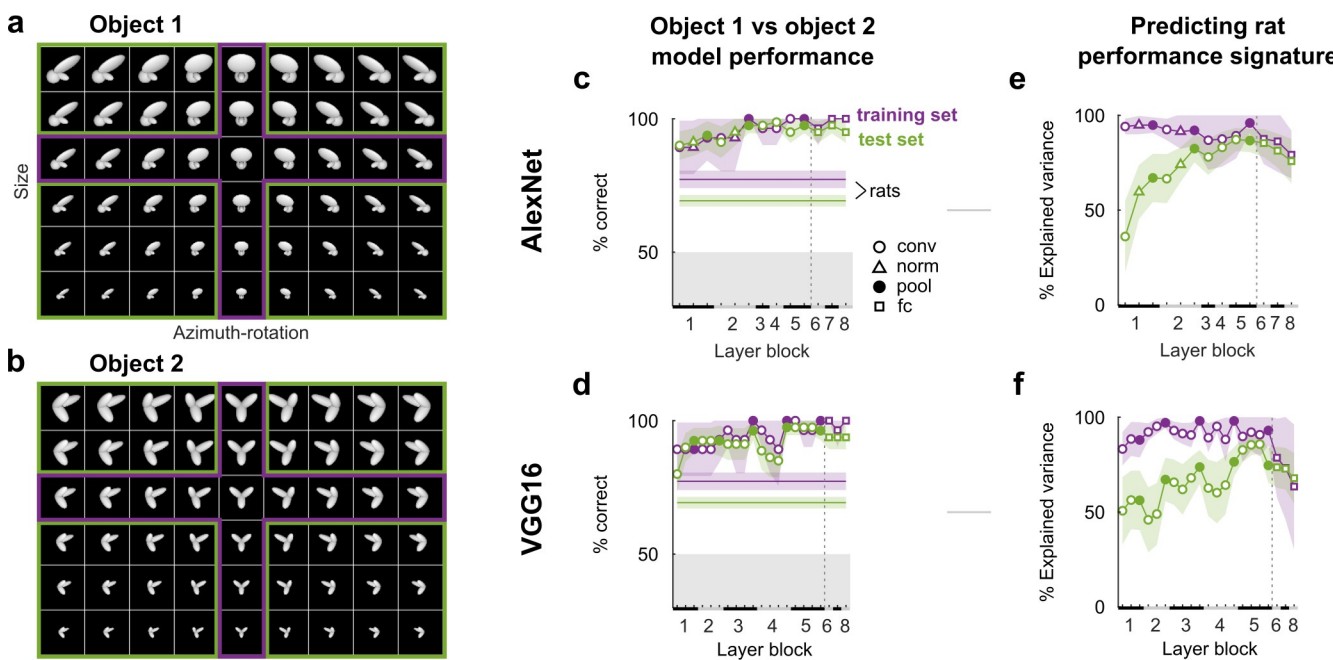

**Fig 2. The first DNN layers account for generalization performance across a range of object transformations, but higher layers explain more variance in the transformation-level behavioral performance signature in Zoccolan et al. [1].** (**a,b**) The full sets of size and azimuth-rotation combinations of the two objects used in the behavioral task. For copyright reasons we do not show the stimuli from the original paper, but images of similar objects created in Blender 2.91 for illustrative purposes only. For the model we did use the originals from [1]. Rats were first trained on a subset of 14 of these transformations (purple) and subsequently asked to generalize to the remaining 40 novel combinations (green). (**c,d**) Average percentage correct discrimination of object 1 and object 2 by models incorporating increasingly more DNN layers of AlexNet (c) and VGG16 (d) (the X-axis indicates the highest DNN layer). Each model's decoder layer was first trained on the same object transformations as the rats, after which performance was evaluated on these trained transformations (purple) as well as the untrained transformations (green). Horizontal lines indicate the average behavioral performance across rats reported by Zoccolan et al. [1]. Black and grey bars on the X-axis indicate layer blocks and markers indicate layer types (see legend insert); the division between convolutional and fully connected layer blocks is indicated by a dashed line. (**e,f**) Percentage variance in performance across stimuli explained by each DNN layer (estimated by a linear mapping fit on the data and stimuli of the training set only) from AlexNet (e) and VGG16 (f). Same conventions as in (c,d). All error bounds are 95% confidence intervals calculated using Jackknife standard error estimates (resampling size and azimuth-rotation combinations).

presented and the rat had to indicate the object identity by licking either a left or right feeding tube. After training, the rats were tested on the full stimulus set, which included novel combinations of size and azimuth-rotation (**Fig 2**A and 2B).

In the original experiment, the rats generalized remarkably well to the object transformations they had never seen before. We trained DNN-based models on the same task (Materials and Methods, Computational modeling, Modeling behavioral tasks) and found that this level of generalization turned out to require surprisingly little processing: the models incorporating only the first convolutional layer of both AlexNet and VGG16 already achieved near perfect generalization performance on the test set (**Fig 2**C and 2D). Even a model trained on a pixel representation achieved 93% correct on the training set and 89% on the test set. Thus, no non-linear processing is required to explain a high level of generalization performance from the trained to untrained object transformations.

The performance of the rats was not the same for all combinations of size and azimuth-rotation: performance was highest for the combination of the most common viewpoint and size in the training set (i.e. the center object in the purple "cross" in **Fig 2**A and 2B), and decreased for objects that deviated from that combination. We call this pattern of performances across object transformations (or–more generally–across images) the behavioral performance signature. To assess whether DNN representations could capture the behavioral results beyond overall generalization accuracy, we tested whether a linear combination of the

activation patterns in a DNN layer could also accurately predict this behavioral performance signature (Materials and Methods, Computational modeling, Predicting behavioral performance signatures). This mapping from DNN representations to behavioral performance per combination of object size and rotation was fit using the training stimuli and data only. Next, we used this behavioral mapping to predict generalization performances for each of the novel (green) object transformations. We calculated the variance in behavioral performance across object transformations that was explained by the DNN representations. These results show that the first convolutional layer only explained a limited amount of transformation-level variance of ∼35–50%, which then sharply increased and slowly reached a maximum of >80% in higher convolutional layers (**Fig 2**E and 2F). Thus, whereas representations in the first layer of AlexNet and VGG16 can already explain successful generalization to untrained object transformations, higher convolutional layers get increasingly better at explaining transformation-level differences in behavioral performance of rats. None of the networks showed any improvement from incorporating the fully connected layers.

## Earliest DNN layers can account for the stimulus-level performance signature across rendered objects and sizes, also for the best performing rats

To further investigate how well the DNN models can explain object and transformation-level differences in the behavioral performance of rats, we turned to a recent study by Djurdjevic et al. [5]. In this study, rats were trained to discriminate a reference object from 11 distractor objects at different sizes ranging from 15 to 35 degrees of visual angle (**Fig 3**A), using a yes-no task in an operant box with liquid rewards (see Materials and Methods, Behavior, Task paradigms). The goal of the study was to use the variable discrimination performance observed across object conditions to infer the complexity of the rats' perceptual strategy. The authors found that there were "good performers", which performed above chance for all distractors at a size of 30˚, and "poorer performers", which performed below chance for more challenging distractors (e.g., the T-shape highlighted in blue in **Fig 3**A).

Can the behavioral fingerprints of good and poorer performing rats be explained by a difference in readout from the same DNN representation, or is a higher-level representation required to explain the behavior of good performing rats? To investigate this, we calculated how much of the object and size-level variance in behavioral performance could be explained, using the same DNN layer activations for good and poorer performing rats, but fitting a separate linear mapping for the two groups as defined before (using the data that was displayed in Figs 1 and 2 of Djurdjevic et al. [5]; Materials and Methods, Computational modeling, Predicting behavioral performance signatures). For both good and poorer performing rats, the percentage explained variance was with ∼85% already near maximum for the first convolutional layer, with an absolute maximum of 91% at pool2 of AlexNet (**Fig 3**B and 3C). Surprisingly, no higher-level representation was required to capture the behavioral performance signature of good performers, suggesting that in principle the difference in behavioral performance signature between good and poorer performing rats can be explained by a different weighting of features of the same representation from the same DNN, in contrast with the idea that good performers relied on more advanced processing of shape information [5]. In sum, object and size-level differences in behavioral performance of both good and poorer performers could be captured well by the earliest layers with the least amount of processing, suggesting that the rats' perceptual strategies could have relied on relatively low-level visual representations.

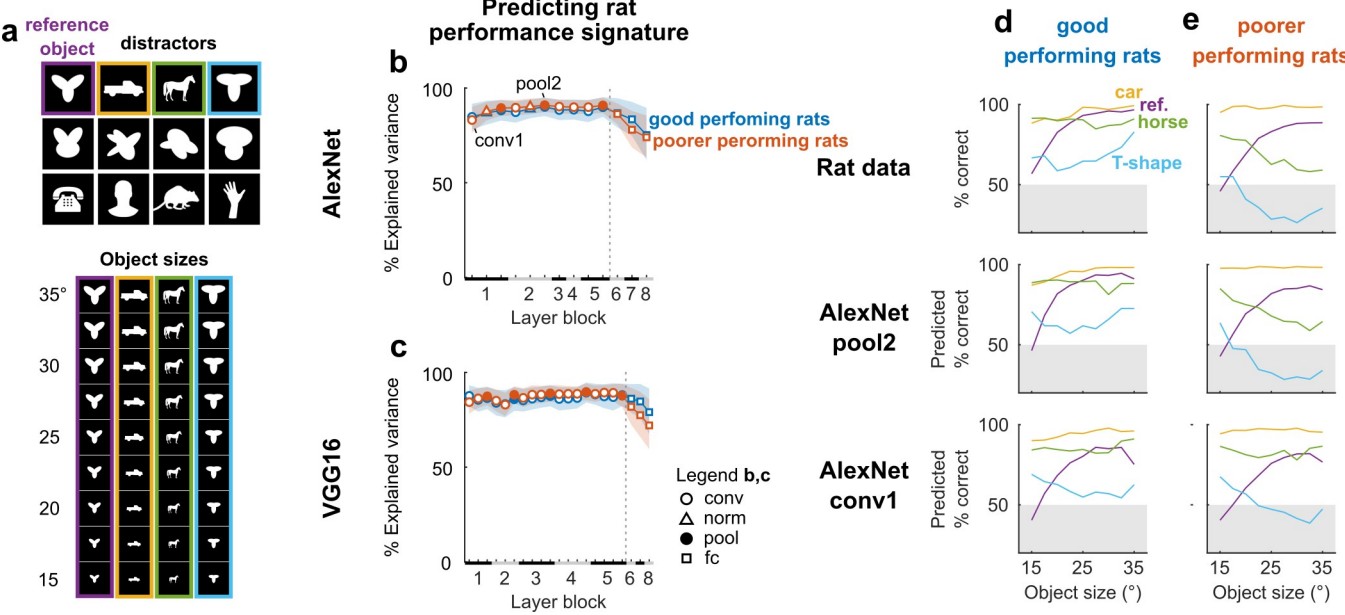

**Fig 3. Stimulus-level discrimination performance signatures for objects from Djurdjevic et al. [5] map onto earliest DNN layers. (a)** Top: the reference object (purple) and 11 distractor objects (rest) that rats were trained to discriminate in the behavioral task. For copyright reasons we do not show the stimuli from the original paper, but similar silhouette images were redrawn by hand or adapted from close-matching clip-art (https://openclipart.org/) for illustrative purposes only. For the model we did use the originals from [5]. The 3 example distractors of Fig 1 in Djurdjevic et al. [5] are highlighted in yellow, green, and light blue. Bottom: in a later phase of the experiment, the rats were trained to tolerate size changes in all objects from 15˚ to 35˚ of visual angle (here only shown for the subset of 4 objects indicated in color on the top). **(b,c)** Percentage of variance in average discrimination performance of good performing (blue) and poorer performing (red) rats, explained by each DNN layer (estimated by a linear mapping with leave-one-out cross-validation) from AlexNet (b) and VGG16 (c). The linear mapping was estimated using the data available in Figs 1 and 2 of Djurdjevic et al. [5]: all sizes of the 4 example objects indicated by purple, yellow, green, and light blue in (a), and the 30˚ size for the remaining eight objects. Black and grey bars on the X-axis indicate layer blocks and markers indicate layer types (see legend insert); the division between convolutional and fully connected layer blocks is indicated by a dashed line. Error bounds are 95% confidence intervals calculated using Jackknife standard error estimates (resampling stimuli). **(d)** Top: average discrimination performances of good performing rats, as a function of object size, for the reference object and the 3 example distractors. Middle: average predicted discrimination performances of good performing rats, based on pool2 of AlexNet, which explained the most out-of-sample variance in behavioral performance. Bottom: same as middle, but for conv1 of AlexNet. **(e)** Same as (d), but for poorer performing rats.

## Mid-level DNN layers can account for natural video categorization behavior

Up to this point, we have only discussed studies that used a small number of computer-renderings of abstract and more naturalistic objects. However, rats have also been shown to be able to learn category rules from more complex natural videos that generalize to novel category exemplars [6]. In this study, rats were trained on a two-alternative forced choice task in a visual water maze (see Materials and Methods, Behavior, Task paradigms) to classify five-second videos featuring a rat (target category), from phase scrambled versions of the target videos and target-matched natural distractor videos featuring various moving objects. The videos were 24 degrees of visual angle as seen from the choice point where the maze splits in two arms [6]. In each trial the target and distractor were presented simultaneously, with one video on the left and the other on the right. The rats were first trained on a training set of 15 videos (**Fig 4**A), initially with a fixed target-distractor pairing, followed by a phase where all possible target-distractor combinations were presented. In the subsequent test phase the rats were probed with 40 novel videos with a fixed target-distractor pairing (**Fig 4**B), without negative feedback for incorrect trials.

The rats were able to generalize well to the novel videos, independently of temporal information or local luminance cues (although local luminance did explain some of the stimulus-level response variance [6]). To test whether this generalization could be explained by other low-level visual information, we trained DNN-based models incorporating increasingly more layers from AlexNet, VGG16, and VGG11-C3D on the same two alternative forced choice task. VGG11-C3D is a convolutional neural network with 3D spatio-temporal filters (16-frame temporal bins) and thus able to also encode temporal features [33]. The features encoded by VGG11-C3D range from moving edges/blobs and changes in orientation or color, to more complex motion patterns such as moving circular objects or biking-like motions [33]. For training the DNN-based models, we took the first 144 frames of the total of 150 frames (4.8 out of 5s) of each video and split those into nine 16-frame bins. The inputs to each classifier were the time-averaged activations of each 16-frame video clip (Materials and Methods, Computational modeling, Feature extraction). All three networks performed very similarly, suggesting there was no real advantage of the motion features encoded by VGG11-C3D for this task. For scrambled distractors the models based on the first layers already performed almost at the behavioral level of rats (**Fig 4**D), but for natural distractors they were at chance and did not exceed rat-level performance until the model included conv4 of AlexNet, conv4b of VGG16, or conv3b of VGG11-C3D (**Fig 4**C). A model trained on a pixel representation achieved only 60% for scrambled distractors and 66% for natural distractors on the test set.

We again tested whether a linear combination of DNN activation patterns could accurately predict differences in behavioral performance across target and distractor frame bins, by fitting a regression model on the average rat performances for each target-distractor combination of the training set and then using this linear mapping to predict test set generalization performances (Materials and Methods, Computational modeling, Predicting behavioral performance signatures). Around 50–65% of variance in behavioral performance for scrambled distractors (**Fig 4**F) and around 40–50% for natural distractors (**Fig 4**E) of the test set was explained by middle convolutional DNN layers.

The results show that stimulus representations of higher DNN layers are required to achieve generalization performance levels on the rat/non-rat classification task that are comparable to rats. Could this layer effect merely be a consequence of the hierarchical architecture of the DNNs, which have properties that systematically change across layers? Even without training, DNNs can provide a powerful set of features, and trends across layers in the ability to capture experimental data can be explained by the number of linear-nonlinear stages rather than the task-optimization of the network [34]. To test this, we repeated our simulations of the behavioral experiments using randomly initialized DNN models (**S1 Fig**). Random networks achieved high generalization in the object classification task of **Fig 2** and the video classification task of **Fig 4** with scrambled distractors but not with natural distractors, where performance levels stayed below those of rats even for the highest layers. Thus, the layer-wise increase in generalization performance in **Fig 4**C depends on the increasingly more abstract features that were learned when the network was trained on image classification.

Together, these results suggest that, in terms of DNNs trained on object recognition, a learned, mid-level representation based on several convolutional and pooling operations is required to explain the observed generalization to novel category exemplars in a classification task of natural videos, while a modest amount of target-distractor-level variance is explained by early to mid-level layers. Again there is no evidence of an added benefit of fully connected layers.

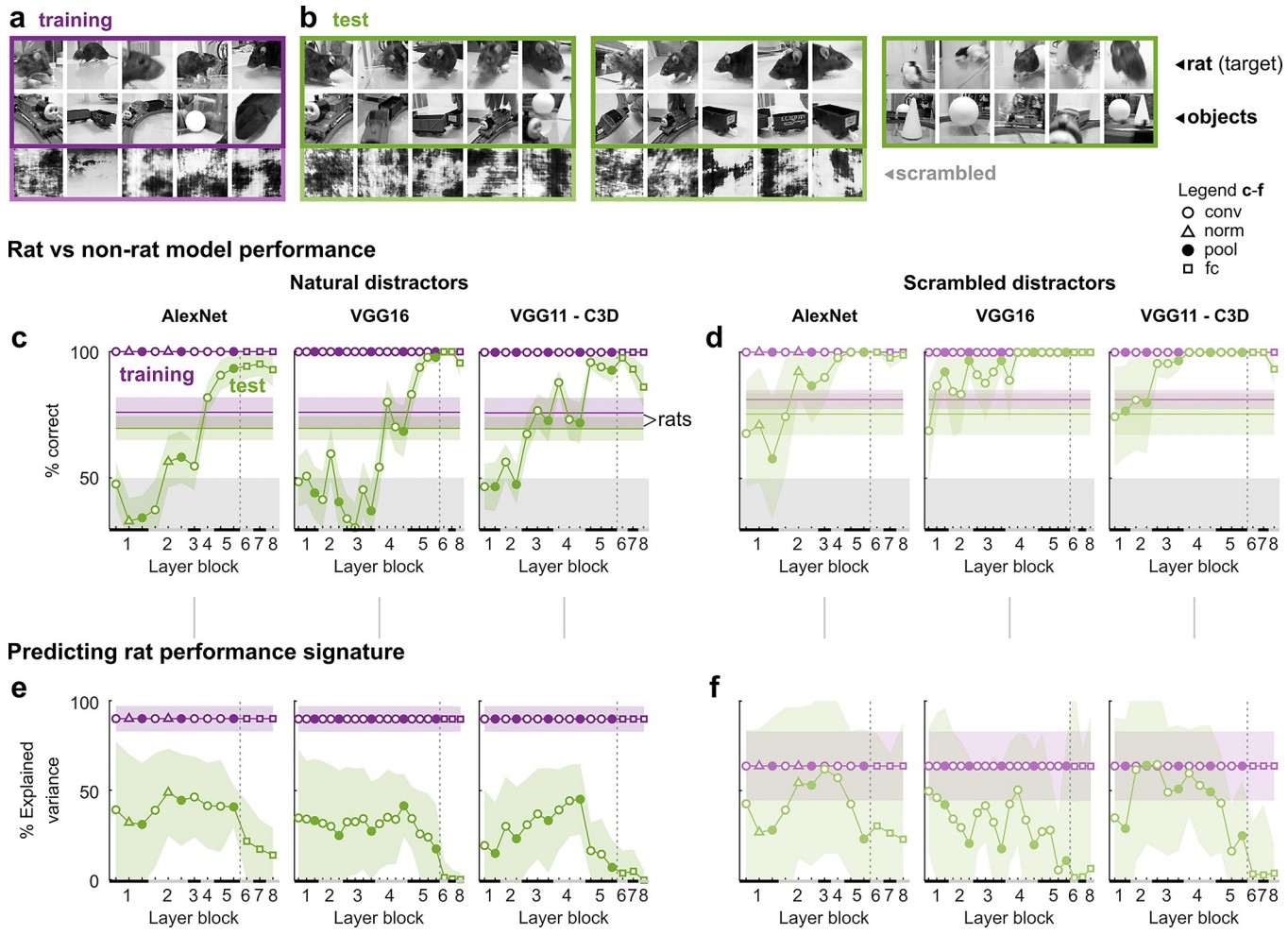

**Fig 4. Mid-level convolutional layers are required for generalization in a rat versus non-rat categorization task of natural videos in Vinken et al. [6].** (**a,b**) Single frames of all the training (a) and test (b) videos that rats were asked to classify in the behavioral task. Each rat was trained with a subset of 15 videos (purple), and tested for generalization with 40 novel videos (green). Ten test videos (the 5 pairs with natural distractor in the left-most green rectangle) were modified to further probe the rats, for example by reducing playback speed to 25% or equalizing average pixel values in the lower-half of the videos [see 6]. (**c,d**) Average percentage correct classification of rat versus non-rat frame bin pairs by models incorporating increasingly more DNN layers from AlexNet, VGG16, and VGG11-C3D (the X-axis indicates the highest DNN layer) and for natural (c) and scrambled (d) distractors separately. Performance is evaluated on the training set (purple; all 50 target-distractor combinations) as well as the test set (green; the 25 tested target-distractor pairs, with 25% playback speed and pixel value modifications for 5 pairs [see (b)]). Black and grey bars on the X-axis indicate layer blocks and markers indicate layer types (see legend insert); the division between convolutional and fully connected layer blocks is indicated by a dashed line. Horizontal lines indicate the average behavioral performance across rats. (**e,f**) Percentage variance in performance across target-distractor pairs explained by each DNN layer (estimated by a linear mapping fit on the training set data and stimuli only) from AlexNet, VGG16, and VGG11-C3D (left to right) and for natural (e) and scrambled (f) distractors. Same conventions as in as in c,d. All error bounds are 95% confidence intervals calculated using Jackknife standard error estimates (resampling target-distractor pairs).

## Representations of natural and scrambled videos in the rat lateral extrastriate cortex change in parallel with representations in DNNs

DNN-based models suggest that several early and intermediate layers of hierarchical processing in an object recognition model are required for a representation that can support natural video categorization. The rodent cortex houses a complex network of specialized higher-order visual areas [35], but is there any evidence for such an intermediate representation in the rat brain? A likely candidate pathway is found in the lateral extrastriate cortex, which anatomically resembles the primate ventral visual stream [36] and shows several functional properties

thought to be typical of an object recognition pathway, such as increased tolerance for changes in position [7], size, rotation, and illumination [8].

Previously, we investigated neural representations of natural and scrambled videos along the putative rodent ventral stream [37]. In this experiment, we presented the 10 natural videos of the training set in **Fig 4**A and their 10 scrambled counterparts in randomized order to awake, passively watching rats which were never trained with these videos. The videos were shown at sizes ranging from 50 to 74 degrees of visual angle (as the eye-to-stimulus distance varied according to the position on the screen, which was optimized for each recording site's receptive field location) separated by a 2 s blank screen, with 10 repetitions per video. We recorded single and multi-unit spiking activity in primary visual cortex (V1), a middle latero-intermediate area (LI), and the most lateral temporal occipital cortex (TO). In the original paper, we reported evidence for an increased dissociation of natural and scrambled videos from V1 to TO, but not for a categorical representation [37].

Here, we investigated the similarity between neural representations of these videos in the putative rat ventral stream and the DNN representations in each layer of each model. We calculated representational dissimilarity matrices (RDMs) based on the neural responses per 16-frame time bin (**Fig 5**A–**5C**, 180 time bins in total), and compared these with RDMs based on DNN layer activity (**Fig 5**D–**5F**). Visual inspection reveals that, similar to the neural RDMs, the DNN layer RDMs show a progression towards an increased dissociation between natural and scrambled videos. However, unlike the neural data, the RDMs of the last fc8 layers suggest a category-like representation differentiating the rat from the non-rat videos (see checkered pattern in the left-upper quadrant).

A further quantitative comparison between each neural and DNN layer RDM showed that the similarity between neural and DNN layer representations increases for LI and TO in later layers, well beyond the similarity for V1 data (non overlapping 95% confidence intervals, **Fig 5G–5I**). When we visualized all between-RDM similarities using multidimensional scaling, the plots suggested a progression from V1 to TO parallel to the progression across successive DNN layers (**Fig 5J–5L**). On average, the distance between the neural and DNN layer RDMs was 2.27 times the average distance between DNN layer RDMs from the three different DNN models. The representational similarity was consistently higher for TO, peaking between pool1 and fc6 for AlexNet (**Fig 5H**), pool2 and conv5a for VGG16 (**Fig 5H**), and pool3 and conv5a for VGG11-C3D (**Fig 5I**). The strong interaction between DNN layer and cortical area could not be explained by randomly initialized DNNs (**S3A–S3C Fig**), suggesting a significant role of the specific features that were learned when the network was trained on image classification.

Overall, these results suggest that the neural representations of the videos in the most lateral visual area TO correspond best to the mid-level representations in DNNs trained on object recognition.

## Discussion

We examined the object recognition capabilities of the rodent visual system, by focusing on several studies and formally assessing the level of abstraction required to explain behavioral performance. Using convolutional neural networks, we assessed at which stage of processing each network can solve the task, to shed light on the extent to which successful generalization performance of rats can be taken as evidence for invariant visual object recognition. This computational approach provides a generally consistent picture of the representations that underlie the behavior of rats: generalization in all tasks could be captured by convolutional layers only, even the most challenging task which required later convolutional layers (**Fig 4**). This

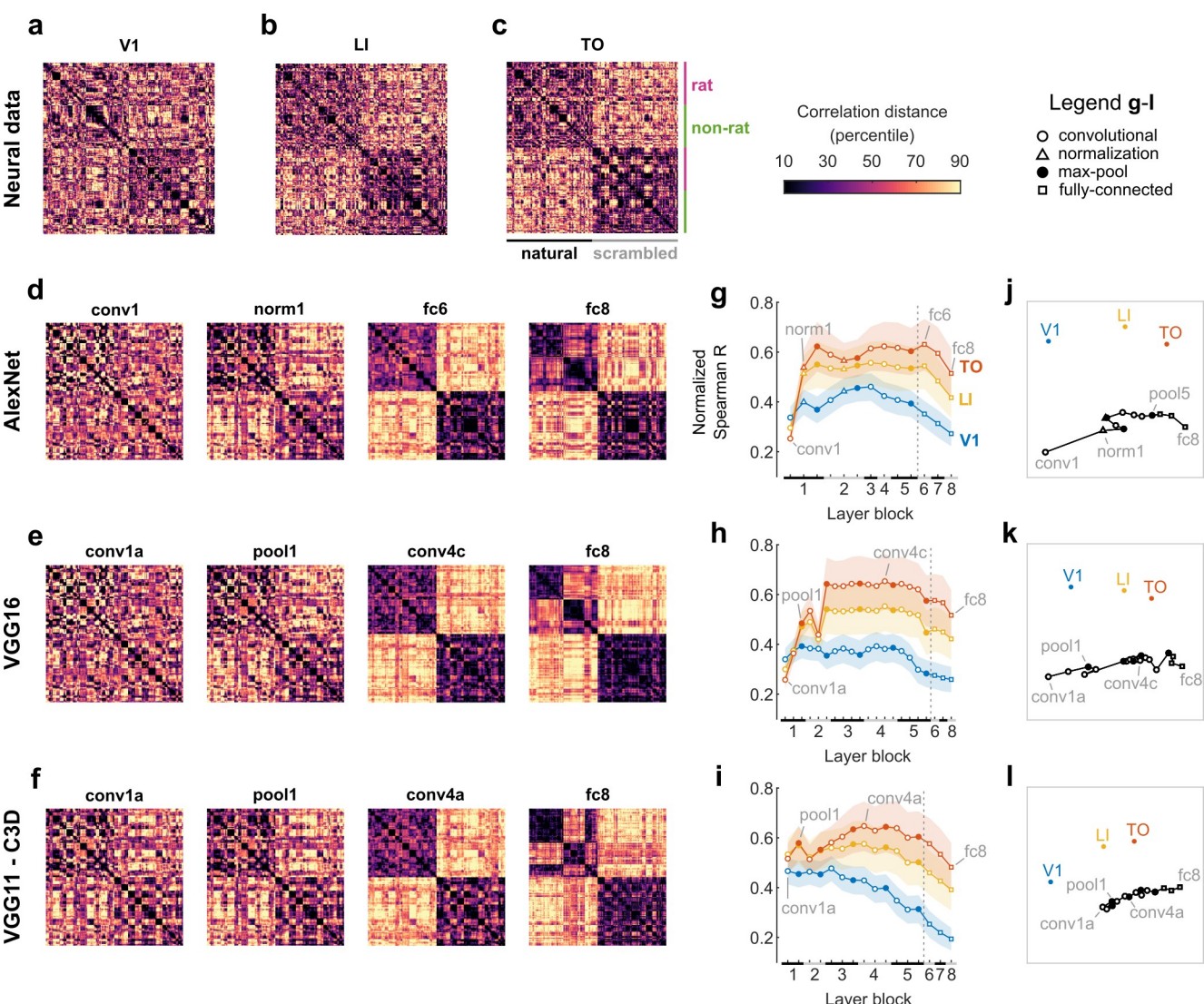

**Fig 5. Neural representational dissimilarity matrices for natural and scrambled videos per putative ventral stream area change in parallel with DNN layer representations.** (**a-c**) Neural RDMs for V1, LI, and TO data. Rows and columns correspond to 16-frame bins: nine per video, with first the five rat videos, then the five non-rat videos, and finally the scrambled versions of the natural videos in the same order. The color scale corresponds to percentiles of each RDM's Pearson correlation distances (excluding the diagonal values). (**d-f**) Artificial neural network RDMs (Pearson correlation distance) for 4 layers of AlexNet, VGG16, and VGG11-C3D: the first layer, the earliest layer for which the RDM corresponds better to extra-striate (LI/TO) data, the layer with the best normalized correlation with neural data (TO in all three cases), and the last layer (fc8). (**g-i**) Spearman correlations between each artificial neural network layer RDM and each neural RDM (calculated using above diagonal elements only), normalized by each area's noise ceiling (V1 in blue, LI in yellow, TO in red). Black and grey bars on the X-axis indicate layer blocks and markers indicate layer types (see legend insert); the division between convolutional and fully connected layer blocks is indicated by a dashed line. Grey text labels indicate the DNN RDMs shown in (d-f). Error bounds are 95% confidence intervals calculated using Jackknife standard error estimates (resampling neural units). (**j-l**) Two-dimensional representation of similarities between neural and artificial neural network RDMs, derived from applying non-metric multidimensional scaling on Spearman correlation distances between RDMs. Each marker corresponds to an RDM and similar RDMs are plotted closer together. Text labels indicate the neural RDMs and the DNN RDMs shown in (d-f).

picture is much more precise than in earlier papers, which generally relied on the assumption that rats could not have used low-level representations to generalize across variations in appearance and identity preserving transformations; an assumption that up until now had not been tested scientifically. As such, this computational approach provides a deeper and more principled understanding of the combined behavioral and neural data available in the literature.

Besides the general performance, there were interesting commonalities between behavior and DNN representations at a more fine-grained level. Despite the ability of the first convolutional layer to generalize across different sizes and rotations in Zoccolan et al. [1], later convolutional layers could explain more transformation-level variance in performance (**Fig 2**). On the other hand, performance differences between objects and sizes in Djurdjevic et al. [5] were equally well explained by the earliest convolutional layers, also for the best performing rats (**Fig 3**). For the natural movies of Vinken et al. [6] the stimulus-pair-level explained variance was generally low, which is more consistent with the finding that the same DNNs fail to account for the image-level behavioral signatures of primates [27].

Later convolutional layers matched the representational geometry increasingly better for extrastriate areas in the rat visual cortex (**Fig 5**), which is consistent with our finding in **Fig 4** that only models that also include later layers generalized to novel exemplars from Vinken et al. [6]. These findings might seem to contradict Cadena et al. [34], who did not find evidence for a hierarchical correspondence between mouse visual areas and layers in a similar DNN. However, Cadena et al. [34] targeted three extrastriate areas that all border V1 and are moderately (lateromedial, LM) to strongly (anterolateral, AL; rostrolateral, RL) associated with the dorsal stream. Both AL and RL show conclusive anatomical and functional dorsal stream properties [36,38]. LM is a gateway to the putative ventral stream, but its role is considered divided because of dense connections to both ventral and dorsal areas [36,39] and inconsistent evidence for ventral-like computations [38,40]. Therefore, it is no surprise that these areas do not map onto the hierarchy of a ventral-like DNN. In contrast, we targeted extrastriate areas LI and TO, which are both positioned laterally to LM and span together with V1 the extent of the putative rodent ventral stream [7]. Being one step downstream from LM, LI lies at the center of this pathway and is both functionally and anatomically considered ventral-like [36,38,39]. Thus, our findings are consistent with functional and anatomical evidence and do not contradict the results of [34].

While our results show that generalization in rodent object vision experiments can be explained by representations in convolutional layers, in primates fully connected layers of the same or similar DNNs capture perceived shape similarity better [18,23,41]. This suggests that the behavioral studies in rats provide evidence for object representations which are markedly less abstract than primates. For the similarity between neural representations in primate inferotemporal cortex and DNN layers the story is less consistent: in some cases it does peak at the fully connected layers [22,42], whereas in other studies it peaks at the last convolutional layer [16,18,43,44], suggesting that it might depend on the particular stimulus set, how anterior in the ventral stream the neural data were recorded [22], and/or the network architecture [45]. This shows the road ahead for future studies on comparing behavior and neurophysiology between animal species: preferably the exact same stimuli and tasks would be used, designed to be able to differentiate among representations and strategies of varying complexity. Currently no such data are available.

It is important to note that the DNNs used in this study are supervised, feedforward neural networks trained on object recognition, which may constrain the features that are encoded by each layer. These models were not designed to model rodent vision [although the overall conclusions did hold when the stimuli were resized so that receptive field sizes in early DNN layers match V1 receptive field sizes (**S2** and **S3D–S3F Figs**)], because they are neither optimized for the same goals, nor are they subject to the same biological constraints of either the rodent or primate visual systems. Therefore, we do not claim that feedforward DNNs trained on object recognition are optimal models of the putative rodent ventral stream, which potentially encodes different information compared to the primate ventral stream. On the other hand, rodents do rely on vision for navigation, which demands to some extent properties that are

fundamental to object recognition, such as tolerance for changes in view-point and illumination [46]. Regardless, DNNs are mechanistic models that can capture the steps of information processing required to solve visual recognition of the main object in natural images. While these steps of information processing in DNNs do not capture all aspects of primate vision [24,25], on a macroscale they do map onto successive stages of primate ventral stream processing [15,16,31]. Thus, as a general modeling framework for object vision, DNNs allowed us to computationally evaluate previous assumptions about the level of abstraction required for generalization in behavioral experiments.

Our results provide important qualifications for researcher assumptions about the stimulus set or task complexity. First, in Zoccolan et al. [1], the overall performance level is surprisingly easy to explain based on activations in the first convolutional layer. A more in-depth look at the behavioral performance signature across object transformations turned out to be more relevant for showing at least mid-level processing. Second, in Djurdjevic et al. [5], the behavioral signature for good performers is actually consistent with early convolutional representations, contrary to the intuitive assumption of the authors that these animals must rely on a complex strategy. Third, a paradigm where only a single stimulus is presented on every trial [as in 1,5] has been suggested [9] to probe more complex strategies compared to a two-alternative forced choice task where the target and distractor can be directly compared [6]. However, here we show that, if anything, the latter paradigm can provide evidence for representations that are at least as high level as the paradigm used by Zoccolan et al. [1].

These considerations are of critical importance for future research on rodent vision. For example, one of the most large-scale initiatives to investigate the visual system in rodents is headed by the Allen Institute, where the most recent efforts yielded nearly 100,000 recorded neurons [47,48]. The interpretation of the neural data is greatly facilitated when behavioral read-outs are available. The current study points to two challenges in this respect. First, we need a computational approach to validate assumptions about the difficulty of the visual tasks and the strategies used for task performance. Second, prior to starting data collection, a computational approach would be highly valuable at the stage of deciding which stimuli to use. Even for the large scale neural recordings [48], the bottleneck for comprehensively characterizing higher-level visual processing could be in the design of the stimulus set rather than the number of recorded neurons.

Finally, it is important to emphasize that the fact that we have not found any evidence for truly high-level visual object recognition behavior does not imply that it is not there or cannot be there. For example, a high generalization performance in the video categorization experiment required higher layer representations than the other experiments, underlining that the evidence we have is not only limited by the capabilities of the rat visual system, but also by the nature of the task. It is possible that none of the studies discussed here really pushed the limits of rodent object vision. A road ahead for addressing this question is to use DNNs to construct stimulus sets and design paradigms that explore the boundaries of rodent vision by getting the best out of them. This will likely also include considerations about the ecological validity of the tasks from the perspective of rodents [34,46,49].

Another interesting avenue is the use of paradigms that investigate visual strategies by means of experimental manipulations that also made the task more challenging and exclude some of the simplest pixel-based strategies. For example, Alemi-Neissi et al. [3] presented stimuli covered by masks that randomly occluded parts of the objects, which effectively excludes the possibility of the animals using one highly specific local-luminance based strategy.

In addition, in a second part of their study, Djurdjevic et al. [5] presented random variations of the reference image to infer perceptual templates, showing that a template-matching model using only one fixed perceptual template could not account for the animal's

performance across image manipulations such as changes in object size. Given the difficulty to relate these strategies to the level of abstraction of the underlying representations, it will be interesting to combine a more computational DNN approach with these template paradigms.

In summary, we used convolutional deep neural networks for a comprehensive and quantitative assessment of the level of abstraction required to explain rodent visual object recognition. A combination of behavioral and neural results reveals a level of invariance comparable to mid-level, classification-trained DNN representations, consistent with the idea of a visual system that is reasonably advanced but not the primary modality. The main conclusion can be phrased in different ways, depending on whether one takes a glass half-full or half-empty perspective. On the one hand, our findings confirm that rodent visual task performance is non-trivial, displays a certain degree of invariance, and requires multi-layer networks to be simulated. On the other hand, the behavioral performance as well as neural responses point to representations of a limited level of abstraction relative to primate vision.

## Materials and methods

### Behavior

**Task paradigms.** In the studies of Zoccolan et al. [1] and Djurdjevic et al. [5], rats were trained on a yes-no task in an operant box with liquid rewards. Briefly, a rat could initiate a trial by licking a central sensor, starting the presentation of a single stimulus with a default presentation time of 3s. Only one object was presented per trial, and the rats had to associate each object identity with a left or right feeding tube. A correct response was given by licking the feeding tube associated with the presented stimulus, which prompted the delivery of the liquid reward.

In the study of Vinken et al. [6], rats were trained on a two-alternative forced choice task (2AFC) in a V-shaped visual water maze [50], with a target video presented at the end of one arm of the maze, and a distractor video at the end of the other arm. A trial was started by placing the rat in the water at a start position where both stimuli were visible. During training, a platform was located only under the target video, which the rat had to reach in order to escape the water maze and end the trial. Both videos were played repeatedly until the trial ended. In the subsequent test phase, a platform was placed under both the target and distractor for trials with novel stimuli, to exclude additional learning from negative feedback for incorrect trials.

For further methodological details about the experimental setups and the tasks, we refer to the original papers.

**Data and stimulus extraction.** We extracted the data and stimuli from the manuscripts of Zoccolan et al. [1] and Djurdjevic et al. [5]. The object images were directly copied from Fig 2A and Fig 1A, respectively. The behavioral data were copied from the values displayed in Fig 2B of Zoccolan et al. [1] (percentage correct based on 70–90 trials for each size and azimuth-rotation and per animal, averaged across $N = 6$ rats), and extracted from Fig 1C and 1D of Djurdjevic et al. [5] using WebPlotDigitizer 4.2 (https://apps.automeris.io/wpd/; percentage correct based on an unspecified number of trials for each size of the target object and of three distractor objects, as well as for the 30˚ size of all remaining distractor objects, for each of $N = 6$ rats). For the analyses of the natural video categorization experiment, we used the original movies from Vinken et al. [6], and had access to the original data which are available at http://doi.org/10.17605/OSF.IO/4W39D (percentage correct based on 14–49 [$M = 30$, $SD = 10$] trials for each target-distractor combination pooled across $N = 5$ rats).

### Neurophysiology

**Data.** The neurophysiological data are from a study published previously in Vinken et al. [37] ($N = 7$ rats) and consist of single cell and multi-unit responses to natural videos recorded

from primary visual cortex (*N* = 50 single cells, *N* = 25 multi-unit sites), latero-intermediate visual area (*N* = 53 single cells, *N* = 33 multi-unit sites), and temporal occipital cortex (*N* = 52 single cells, *N* = 26 multi-unit sites). For methodological details about the experimental setup and recordings, we refer to the original paper.

**Representational dissimilarity matrices.** For comparison with DNNs (see Computational modeling, Comparing neural and DNN stimulus representations), we calculated neural representational dissimilarity matrices [RDMs; 51]. In short, for each of the 20 five-second videos we considered the first nine 16-frame bins (533ms each–together covering 144 out of the full length of 150 video frames, or 4.8 out of 5 seconds) to match the temporal bins explained under Computational modeling. This yielded a total of 180 16-frame bins. Next, for each of these 16-frame video bins, we calculated a neural response vector with the average standardized firing rate of each single and multi-unit response, taking into account a temporal shift corresponding to the response latency which was estimated separately for each neuron or site [see 37]. This resulted in 180 response vectors (one per 16-frame bin of each stimulus), which were correlated pairs-wise (Pearson *r*) in order to obtain RDMs with distances $1-r$. Stimulus pairs that elicit a similar neural response pattern result in a lower dissimilarity.

## Computational modeling

**DNNs.** We used three trained DNN architectures as computational models of visual processing in the primate ventral stream: AlexNet, VGG16, and VGG11-C3D. AlexNet [52] and VGG16 [53] were both taken from the MATLAB 2017b Deep Learning Toolbox and had been pre-trained on the ImageNet dataset [32] to classify images into 1000 object categories. Both architectures have been extensively used to model ventral stream processing and object perception [14,16–20,22,23]. For the experiments involving videos we also used VGG11-C3D [called C3D in 33], an architecture which is similar to VGG11 [53], but performs convolution and pooling across the two spatial dimensions *and* a temporal dimension (operating on 16-frame bins). This network had been pre-trained on the Sports-1M dataset [54] to classify videos into 487 sports categories and has previously been used to model human brain responses to natural videos [55]. All three networks consist of a sequence of convolutional and max pooling layers followed by three fully connected layers. Only AlexNet also includes local response normalization layers. Each convolutional and all except the last fully connected layers were followed by a rectifying linear activation function (ReLU). We always extracted activations of individual units from each layer before the ReLU. For each model the full set of layers is typically divided in eight layer blocks (see **Table 1**).

**Table 1. Nomenclature used to refer to each architecture's layers and the division across layer blocks.**

| Layer block | AlexNet | VGG16 | VGG11-C3D |
|---|---|---|---|
| 1 | conv1 norm1 pool1 | conv1a, b pool1 | conv1a pool1 |
| 2 | conv2 norm2 pool2 | conv2a, b pool2 | conv2a pool2 |
| 3 | conv3 | conv3a, b, c pool3 | conv3a, b pool3 |
| 4 | conv4 | conv4a, b, c pool4 | conv4a, b pool4 |
| 5 | conv5 pool5 | conv5a, b, c pool5 | conv5a, b pool5 |
| 6 | fc6 | fc6 | fc6 |
| 7 | fc7 | fc7 | fc7 |
| 8 | fc8 | fc8 | fc8 |

*conv*: convolutional layer, suffixes a, b, or c are used if a block contains multiple conv layers; *norm*: local response normalization; *pool*: max pooling operation; *fc*: fully connected layer.

**Feature extraction.**   Each network has learned a rich series of feature representations which can be accessed from every layer by obtaining unit activations for an input stimulus. We calculated these activations for every convolutional, normalization, max pooling, and fully connected layer, standardized the values across inputs and reduced the dimensionality using principal component analysis (always retaining the full set of principle components). In the case of videos, activations were averaged over time within each 16-frame bin before dimensionality reduction using principal component analysis. For VGG11-C3D this was done by taking the mean activation of each spatiotemporal convolutional kernel across the temporal dimension. For AlexNet and VGG16 (which were pre-trained on static images) we averaged outputs across frames for each 16-frame bin, by taking the mean activation of each unit across frames.

Note that the parameters for the standardization and the transformation to principal component space were always calculated only based on the stimulus set that the animals were trained on in the behavioral experiments, thus excluding the stimuli of the test set (except for the Djurdjevic et al. [5] experiments, where there was no separate test set for generalization). The resulting feature vectors were then used to model the behavioral task for comparison with neural representations.

**Modeling behavioral tasks.**   Visual object recognition experiments were modeled using pre-trained DNNs, by placing a new linear decoder layer on top of an original network layer and training the new decoder on the specific task and stimuli of the experiment. To simulate visual systems with lower levels of ventral stream-like processing, we not only trained a decoder on top of the penultimate DNN layer [as in 27,28], but trained separate decoders for each DNN layer, each constituting a different model (**Fig 1**B). For each task, the decoder was a linear support vector machine (SVM) classifier trained on the same binary tasks that the rats were trained on (object 1 vs 2 for [1]; reference object vs distractors for [5]; rat movies versus distractor movies for [6]), using standardized DNN layer activations in principle component space as inputs (note that this is equivalent to training a fully connected neural network layer with binary output). As a baseline control, we also trained a classifier on a pixel representation obtained by transforming pixel values to a principal component space calculated from the stimulus set that the animals were trained on (i.e. identical to what we did with DNN activations). Each classifier was trained on the experiment's training set only, and tested for generalization on the test set to assess whether its feature representation can support the task (i.e., whether the object/category representations are such that a linear combination of features optimized for the training set generalizes to the test set). Each classifier was trained using the MATLAB 2017b function *fitclinear*, with the limited-memory BFGS solver and default regularization. For yes-no tasks with one stimulus per trial [1,5], task performance was evaluated for each stimulus individually: classification was considered correct if the stimulus fell on the correct side of the SVM decision boundary (**Fig 1**D). For the two-alternative forced choice task, which had both a target and distractor per trial [6], a model response was considered correct if the target stimulus fell on the correct side of the distractor, relative to the decision boundary. That is, when (a) the target and distractor were both on their respective correct side of the SVM boundary (**Fig 1**E i), (b) both were on the target side, but the target was further away from the decision boundary (**Fig 1**E ii), or (c) both were on the distractor side, but the target was closer to the decision boundary (**Fig 1**E iii). Code for this model is available at http://doi.org/10.17605/OSF.IO/4W39D.

In the behavioral experiments, rats were not head fixed or fixating, so the actual retinal projection of an object image could vary from trial to trial. Importantly, rats ware trained with this variability and the distributions of retinal projections during training trials should cover those during test trials. We found that explicitly modeling such variability by varying object

positions during training and testing for the Zoccolan et al. [1] experiment did not lead to qualitatively different results, thus we proceeded without such variability.

**Predicting behavioral performance signatures.** The probability of correct response by the rats was not the same for every trial, but varied depending on the presented stimulus (i.e. size and rotation in [1], object and size in [5], and category exemplar in [6]). We call this pattern of percentages correct across training or test images in the task the behavioral performance signature [27]. We used a linear regression approach to assess whether a linear mapping could predict these behavioral performance signatures from activation patterns in DNN layers. Specifically, for each DNN layer, a partial least squares (PLS) regression of the (logit-transformed) average rat performance onto the DNN layer features was fit using stimuli and data from the training phase of the experiment only. For the experiment with a 2AFC task [6] we used the DNN features of the target stimulus minus the DNN features of the distractor stimulus as predictor variables for each stimulus pair. We used three PLS components for the experiment with two objects [1], and ten components otherwise.

The accuracy of each model at predicting behavioral performance signatures was assessed by calculating the percentage explained variance as the squared Pearson correlation between predicted performance values $y^{pred}$ and observed (logit-transformed) performance values $y^{obs}$: $corr(y^{pred}, y^{obs})^2$. For experiments with an independent generalization phase [1,6], the PLS regression was fit using all stimuli and data from the training phase of the experiment, and thus out-of-sample prediction accuracy was assessed on the independent test set consisting of all stimuli and data from the generalization phase. Leave-one-out cross-validation was used instead when there was no independent generalization phase in the experiment (i.e., **Fig 3**).

**Comparing neural and DNN stimulus representations.** In order to estimate how closely the representational geometry of each DNN layer matched that of visual areas along the putative rodent ventral stream, we calculated RDMs based on DNN features. As for the neural data, the stimulus feature vectors (one vector for each 16-frame video bin) were correlated pairs-wise in order to obtain an RDM for each DNN layer. As for neural RDMs, stimulus pairs that share a similar representation across features in a layer result in a lower dissimilarity. We then quantified the correspondence between neural and DNN RDMs by calculating the Spearman correlation between off-diagonal upper halves of the matrices. We normalized the correlations between neural and DNN RDMs by dividing by each area's noise ceiling. To estimate the noise ceiling we split the trials per movie in two halves and computed the Spearman correlation between the two resulting neural RDMs (one from each split half). The noise ceiling was the Spearman-Brown-corrected average (across 1000 random splits) split-half correlation.

## Supporting information

**S1 Fig. Randomly initialized AlexNet can account for the main results of Figs 2 and 3, but not Fig 4.** We repeated the analyses of the main figures with 10 randomly initialized AlexNet architectures (using the same scheme as the trained version, i.e., uniform Glorot initialization for the weights [56] and zero bias). (a,b) the analyses of Fig 2C and 2E. (c) the analysis of Fig 3D. (d,e) the analyses of Fig 4C–4F. All error bounds are 95% confidence intervals calculated using Jackknife standard error estimates (resampling the 10 random initializations). All other conventions match those of the corresponding figures in the main text. (TIF)

**S2 Fig. Main results of Figs 2–4, for stimuli scaled to match receptive field (RF) sizes in early DNN layers to V1 RF sizes.** The V1 RF sizes reported in pigmented rats cover a broad range between 3 and 20+ degrees of visual angle [8,57,58]. To match DNN RF sizes in early layers with the range observed in rat V1, we downsized the stimuli so that the RF sizes in

conv1 (11x11 pixels), pool1 (19x19 pixels), and conv2 (51x51 pixels) corresponded to 5, 8.6, and 23.2 degrees of visual angle, respectively, and repeated the analyses of the main figures in AlexNet. All stimuli were downsized from a default size of 227x227 pixels to match the reported presentation size in degrees of visual angle (followed by padding to the DNN input size of 227x227 pixels). (**a**,**b**) the analyses of **Fig 2**C and 2E, after downsizing the images to 129x129 pixels (57%) to match the largest object size of 40 degrees of visual angle. (**c**) the analysis of **Fig 3**B, after downsizing the images to 99x99 pixels (44%) to match the largest object size of 35 degrees of visual angle. (**d**,**e**) the analyses of **Fig 4**C–4F, after downsizing the videos to 53x53 pixels (23%) to match 24 degrees of visual angle. All conventions match those of the corresponding figures in the main text. The effects of downsizing the stimuli are most notable in (d) and (e), where the stimuli were reduced to a much lower resolution, leading to the model requiring a higher DNN layer to reach rat-level accuracies and explain most stimulus-level variance.
(TIF)

**S3 Fig. Main results of Fig 5, for randomly initialized AlexNet and for stimuli scaled to match RF sizes in early DNN layers to V1 RF sizes.** (**a**-**c**) We repeated the analyses of **Fig 5**D, 5G and 5J with 10 randomly initialized AlexNet architectures (see **S1 Fig**). Error bounds are 95% confidence intervals calculated using Jackknife standard error estimates (resampling the 10 random initializations). (**d**-**f**) The analyses of **Fig 5D, 5G and 5J**, after downsizing the videos to 137x137 pixels (60%) to match 62 degrees of visual angle (see explanation **S2 Fig**). Error bounds are 95% confidence intervals calculated using Jackknife standard error estimates (resampling neural units). All other conventions match those of the corresponding figures in the main text.
(TIF)

## Acknowledgments

We thank Davide Zoccolan and Thomas P. O'Connell for helpful comments on this work.

## Author Contributions

**Conceptualization:** Kasper Vinken, Hans Op de Beeck.

**Formal analysis:** Kasper Vinken.

**Funding acquisition:** Kasper Vinken, Hans Op de Beeck.

**Supervision:** Hans Op de Beeck.

**Visualization:** Kasper Vinken.

**Writing – original draft:** Kasper Vinken.

**Writing – review & editing:** Kasper Vinken, Hans Op de Beeck.

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
