## [Decision Letter · Decision Letter 0]

24 Aug 2020

Dear Dr. Vinken,

Thank you very much for submitting your manuscript "Using deep neural networks to evaluate the complexity of rodent object vision" for consideration at PLOS Computational Biology.

Your manuscript was reviewed by members of the editorial board and three independent reviewers. In light of the reviews (below this email), we would like to invite the resubmission of a significantly-revised version that takes into account the reviewers' comments.

As you will see, one of the main issues raised by the reviewers relates to the claim that this study provides insight into the complexity of rodent vision. Reviewer #1 wonders whether DNNs are really a sound yardstick for this complexity; Reviewer #2 notes that no mathematical definition of "complexity" was provided and wonders whether the complexity arises due to the nature of the task or the structure of the rodent visual system; Finally, Reviewer #3 notes that the paper lacks considerations of alternative explanations for the conclusions that are drawn about the complexity of rodent vision.

In addition, two of the reviewers indicated that a lot of details are missing, which makes parts of the paper hard to understand and raises concerns about reproducibility of the results. Personally, I'm a strong advocate of Open Science, so I hope you will give some consideration to Reviewer #2's suggestion to share your data and code.

In addition to these two highlighted issues, the reviewers raised a number of other issues that should be addressed. For details, I refer to the reviews attached below.

We cannot make any decision about publication until we have seen the revised manuscript and your response to the reviewers' comments. Your revised manuscript is also likely to be sent to reviewers for further evaluation.

Sincerely,

Ronald van den Berg

Associate Editor

PLOS Computational Biology

Wolfgang Einhäuser

Deputy Editor

PLOS Computational Biology

Reviewer's Responses to Questions

**Comments to the Authors:**

Reviewer #1: SUMMARY

The paper presents computational modelling of the tasks performed in three high-profile rodent object vision studies from the existing literature. For each, the authors assess how well the rodents' behavioural task can be performed on the basis of the features at different layers of an object-categorisation-trained feedforward deep convolutional neural network (DCNN).

The text is clearly written, with a cogently explained motivation, well presented figures, and clearly explained methods and results. The work makes clever use of DCNNs to move beyond intuitions about whether a certain behaviour could be performed on the basis of, for example, V1 representations, and instead turn the question into an empirical one: what image-computable set of features can support a linear readout that would enable the behaviours shown by experimental animals? The use of real stimuli and tasks from a range of actual prior high-profile work on object vision in rodents is elegant. The findings strike an important note of caution that apparently "complex" tasks may often be performable on the basis of apparently "simple" features; computationally testing the features required to perform a task is essential. I found the paper to be generally excellent!

MAIN COMMENTS

My only major question is about the motivation and interpretation of the modelling results: Are classification-pretrained DCNNs sound models against which to evaluate the "complexity" of a visual system?

The paper is pitched as a test not of how well particular DCNNs perform as models of rodent vision, but of how "complex" rodent vision is, as indexed by how well aspects of it can be predicted from particular DCNNs. For example, the abstract proposes to test whether "invariant object recognition occurs in rodents", and Figure 1 proposes that the "lowest-level mode [layer] to solve the task indicates task difficulty". The complexity of rodent vision is certainly an interesting question, and I do agree that DCNNs can shed some light on it, in the sense that they are hierarchical models which begin as pixel representations, then move through nonlinear transformations towards object categorisations. If very early layers can well predict rodent brain and behavioural data, this suggests that there is no need to suppose sophisticated invariant object representations in the rodent brain. However, the match or non-match of later layers is harder to interpret, since it depends heavily on the training objective of the network.

The current wording of the paper presents DCNNs as simply moving from "less complex" to "more complex and primate-like" representations, which is an oversimplification. The specific networks used here are pretrained on taks and datasets that have little relation to the natural tasks of either rodent or primate vision (2 models are trained to classify the main object in an image as being one of 1000 object classes, of which around 120 are different dog breeds; the other model is trained to classify sports videos into 487 sports). We know that (feedforward, supervised) DCNNs fail to capture a lot of the "visual intelligence" of primate vision (e.g. Geirhos et al (2020) https://arxiv.org/abs/2004.07780). Despite them being the best currently available models of primate vision, DCNNs fail to be "primate ventral stream like" (implication around lines 100 and 295) in lots of important ways: they are trained on ecologically unrealistic objectives that only capture a tiny portion of what vision does, and perhaps as a result are more fragile, susceptible to noise, reliant on texture cues, etc, than primate vision.

More fundamentally, models of this kind are classifiers, moving from pixels to whatever carving-up of category space they have been trained to perform. The extent to which this leads to the development of "complex" representations (e.g. view invariance) depends more on the training data and task than on the type of model. Although the 1000-class Imagenet task does lead to networks with decent position and view invariance, it also produces networks which over-rely on textures (Geirhos et al (2018) https://arxiv.org/abs/1811.12231), likely because it can be reasonably well solved on the basis of local image fragments (Brendel & Bethge (2019) https://arxiv.org/abs/1904.00760). Models trained on different tasks will learn different image transformations. It is therefore inaccurate to view the layer of a particular DCNN as a straightforward proxy for "complexity" or "difficulty" of a visual task.

All of which is to say: object-classification supervised DCNNs are obviously the best models of high-level vision we currently have, both in terms of their sheer accuracy and their match to primate brain and behavioural data. However, in using them as a yardstick for the "complexity" or "invariance" of another model visual system, it's important to acknowledge that the features in the models used here arose from training on very limited tasks, which likely don't align with the goals of a rodent. It is important to specify that the DCNNs considered here are *supervised feedforward 1000-class object-categorisation networks* - a very different "level of abstraction" relationship would likely hold across the layers in e.g. a generative adversarial network or a variational autoencoder. It would be very interesting to see how an unsupervised DCNN model fared, but I think this point can be adequately covered by changes to the text in the Abstract, Introduction, Figure 1, and perhaps Discussion, without new experiments.

MINOR COMMENTS

- If features are extracted from each layer pre-ReLU, does this mean that the conv1 layer is simply a linear combination of pixels? It might be helpful to see a raw pixel model as a baseline (i.e. linear classifier is trained to perform task directly on pixels of image), since performance is so high for all layers in many plots. e.g. line 122 "we...found that this level of generalisation required surprisingly little processing" - does it require *any* processing, or is this a task that can be trivially solved with no nonlinear image transformations at all?

- In Figure 3, the faint curved lines connecting the network layer labels in 3b to the y-axis titles of 3d look distractingly like confidence bounds on the data plotted in 3b, and took me a while to interpret. May be better removed altogether?

- Around line 285 when interpreting how the layerwise correspondence with rat neural data relates to that found with human/non-human primate brain data, it seems as if most human brain data now point to something fairly similar to the correspondences shown for rats in Figure 5. E.g. peak correspondences are found for mid-to-late layers, falling in the latest layers, in posterior VTC (https://www.nature.com/articles/s41598-020-59175-0), LOT and VOT for several stimulus sets (https://doi.org/10.1101/2020.03.12.989376) and a broad IT swath (https://doi.org/10.1101/2020.05.07.082743).

TYPOS / WORDING

- line 350 / 352: "at one hand"? Not "on one hand"?

- line 358: "rats WERE trained"

Reviewer #2: See Attached Document.

Reviewer #3: Summary

The manuscript "Using deep neural networks to evaluate the complexity of rodent object vision" by predicting behavioral data of rats from three different paradigms from convolutional layers of three different DNNs, and by using a representational dissimilarity analysis between DNN at different layers and rat neurophysiological data from different areas. The authors conclude that rodent vision can be captured by mid-level layers of DNNs, earlier layers than required for primate vision, and thus requires lower complexity.

Major Comments

I have to admit I have some trouble with this paper.

First of all, it could improve on clarity. It's really hard to understand the exact stimulus paradigms from the paper. Yes, they can be found in the orignal submissions, but it would be good to summarize each paradigm in a plot to get the reader on board and not force her/him to look up technical details in other papers. Furthermore, important details seem often hidden behind unnecessary high level language and could easily be made more concise. To give an example: You talk about "behavioral patterns". That could mean anything from ethogram to single trial behavioral responses. I think in the current manuscript it's neither, but I also couldn't find out what it is exactly. Things like that make the manuscript hard to read and understand. Finally, important details and measures are missing. For instance, I couldn't find an exact description of the behavioral data (only that it was extracted from the plots of the original manuscripts). What are the inputs? What's exactly measured? This is important to understand what you are analyzing. In addition, almost all plots are missing any sort of error bars to judge whether the difference (e.g. between different layers) is significant or not.

The major issue that I have is that I have a hard time believing in the conclusions. If I had to rephrase what was done in the paper it would be that midly complex tasks (or rats' behaviors on it) were fitted with very powerful deep networks on likely very few training trials. Because the test error peaks at mid-level layers of these networks, it is suggested that the rat visual system is less complex than primates (because they peak later). However, there are almost no controls or alternative explanations suggested by the authors. To list a few that I would consider:

- The training performance of these networks is basically flat everywhere. This means that the task is likely linearly separable on almost all layers. This means that there is likely also a linear function that can solve the test samples perfectly on almost every layer, you just cannot find it from limited training data. I am positive that if you trained on the test trials as well you would find that function. However, if that's the case, the conclusion changes, because the reason for the peak at a certain layer is not representational complexity but your ability to fit the data on limited trials.

- Predictive performance is probably not a very fine grained score to distinguish between different feature complexities in networks. For me, that's the main conclusion of the Cadena paper you discuss. If you have a complex feature space you can fit many things well, in particular if you have few trials. For that having controls like randomly initialized networks like in Cadena et al. is important. If you run those, make sure to use a batch norm after the final layer to counter possible large differences in scale in the random architecture. Furthermore, you could try to use other meaures to check whether the behavior of the networks is consistent with the behavior. A useful reference might be "Geirhos, R., Meding, K., Wichmann, F. A. (2020). Beyond accuracy: quantifying trial-by-trial behaviour of CNNs and humans by measuring error consistency. arXiv preprint arXiv:2006.16736."

- What about the confounder of task complexity? You mention it in the Discussion but it would be important to control for it.

- What about image scale in RDS? Do the similarity measures change with image scale? Deep networks have a strong correlation between spatial size of a feature and feature complexity. Rats likely have large receptive fields compared to primates (in mice it's definitely true; I don't know rats well enough). So this confounder needs to be controlled for by presenting different scaled input images/videos when computing the RDS.

- Finally, it's known that deep networks use different strategies to solve problems (i.e. they are texture biased. See work by Geirhos and collegues or by Brendel on BagNets). The reason for that is that even a large scale dataset like imagenet can be solved in many ways and texture seems to be easier to learn for those networks. I feel you conclusions rests on the assumption that the networks and the rats solve the task in the same way, but that's likely not the case. How does this affect your conclusion? This needs to be discussed, analyzed, and controlled for.

Minor Comments

- 43: "is not yet proven scientifically. " Please rephrase, empirical insights are not proven.

- Fig 1: "The primate ventral stream transforms visual representations where object manifolds are entangled, into more abstract visual representations where object manifolds are linearly separable." I don’t think that’s definitely shown yet. I’d phrase more carefully.

- 57: You might consider including "S. A. Cadena, G. H. Denfield, E. Y. Walker, L. A. Gatys, A. S. Tolias, M. Bethge, and A. S. EckerDeep convolutional models improve predictions of macaque V1 responses to natural images

PLoS Computational Biology, 2019" in the list of references.

- 250ff: "As such, this computational approach provides an unprecedented mechanistic understanding of the combined behavioral and neural data available in the literature." I don't understand why your work is a mechanistic understanding.

- The entire paper is about rats. I would reflect that in the title/text and change "rodents" to "rats".

- Number of trials for behavioral experiments not listed.

- 423: "always retaining the full set of principle components" I don't understand. You do PCA but keep all PCs? Why not use the original data then?

- Fig3d: What does "predicted correct" mean? Correctly predicted rat performance or correctly predicted trials? If the latter how are good and bad performers defined in the DNN?

- Sec 2.2: Not clear to me what was fitted. Did you train the network to predict the rats' performances or did you train it to solve the task?

- Sec 2.3: Task not clear to me. What did the rats have to distinguish?

- 127ff: Unclear to me what you predicted. Was it the performance of the rat for each transformation?

- Why use representational dissimilarity and not directly predict the responses. See recent paper by Kornblith et al. (https://arxiv.org/abs/1905.00414) on comparing representations of networks.

**Have all data underlying the figures and results presented in the manuscript been provided?**

Reviewer #1: **No: **An OSF repository has been created and authors agree to make available upon publication.

Reviewer #2: **No: **The relevant codes and images should be shared as public repository.

Reviewer #3: **No: **The authors said that they are available, but they have not been provided.

PLOS authors have the option to publish the peer review history of their article (what does this mean?). If published, this will include your full peer review and any attached files.

Reviewer #1: No

Reviewer #2: No

Reviewer #3: No
---

## [Decision Letter · Decision Letter 1]

9 Dec 2020

Dear Dr. Vinken,

We have now received reviews (attached below) of your revised manuscript from the original three Reviewers. As you will see, two of the reviewers are satisfied, but Reviewer #1 is still concerned about your definition/interpretation of task "difficulty" and asks for clarification of this concept in the manuscript. I agree with this comment and would therefore like to invite you to submit another minor revision before I make a final decision.

Sincerely,

Ronald van den Berg

Associate Editor

PLOS Computational Biology

Wolfgang Einhäuser

Deputy Editor

PLOS Computational Biology

[LINK]

Reviewer's Responses to Questions

**Comments to the Authors:**

Reviewer #1: This revision fully addresses the few previous criticisms I had. The change of title removes the potentially-problematic claims about complexity, the inclusion of a randomly-initialised comparison DNN in a Supplementary Analysis is interesting and helps relate this work to other recent explorations of random-network performance in rodents and primates, and the improved detail and clarity in the Methods reporting are welcome. I think this is a strong paper!

Reviewer #2: Most issues were addressed.

Minor points:

1. Figure 3 and related results: I am still unclear on how the authors divide the DNNs into good and poor performers. Did they train separate DNNs?

2. OSF link does not have any software or data.

3. Line 72: "... a models ..."

Reviewer #3: Also uploaded as attachement.

Thank you for your detailed responses, the additional experiment, and the revision of the manuscript. I think the manuscript improved a lot. However, I have to re-emphasize one point here which I think it really crucial for the interpretation of your results (I just copy my point and your response here again).

---

My comment: The training performance of these networks is basically flat everywhere. This means that the task is likely linearly separable on almost all layers. This means that there is likely also a linear function that can solve the test samples perfectly on almost every layer, you just cannot find it from limited training data. I am positive that if you trained on the test trials as well you would find that function. However, if that's the case, the conclusion changes, because the reason for the peak at a certain layer is not representational complexity but your ability to fit the data on limited trials.

Your response: The question is not whether the two classes of stimuli used in the experiments are linearly separable – any set of N randomly labeled, non-collinear points will be linearly separable in ≥N-1 dimensional space, so this point could be made for virtually any behavioral stimulus set given the dimensionality of (early) DNN layers (and visual areas). The critical question is whether a hyperplane optimized for a given training set (i.e. the stimuli used to train rats) will generalize to a new set unseen by the classifier (i.e. the stimuli used to test generalization in rats). Thus our conclusions do not depend on the linear separability of the test set (as this is trivial), but on how well a hyperplane that was optimized for the training set generalizes to the test set.

That being said, we realize “complexity” of representations is not the right phrasing to express the goal of our work here, which is to evaluate task difficulty with DNNs trained on object recognition as a framework to better understand what evidence is contained in previous experiments. For this reason we changed the use of the word complexity in the title and throughout the paper (see also our responses to Reviewers #1 and #2).

---

I don't agree with your response for the following reason: If you talk about "complexity" (now dropped) or "difficulty", you talk about a property of the task based on the given stimuli and possibly the representations in a layer of a deep network. However, you don't want "difficulty" to depend on the learning algorithm of your linear classifier (e.g. the optimization or regularizer) that you use to predict your targets from a particular layer. In my view, you are confounding two things here which should be untangled in the manuscript: (A) linear separability or (B) how characteristic certain features in a layer are for a problem (I'll make that clearer below).

Re (A): Task difficulty could simply mean whether the problem is linearly separable in a given feature space. You seem to refer to this definition as Fig 1a exactly depicts this situation. In particular, this would mean that (given enough trainings data) there is no linear classifier that can separate the two classes in feature space. However, as I pointed out and you agreed, you cannot detect this case since you don't have enough data and your task is likely always linearly separable. However, Fig. 1a is misleading in this respect. Your responses to this point are that (a) all behavioral set suffers from this and (b) that it's about the generalization of a hyperplane learned on a training set to a test set. (a) might be true but I don't think it's a valid counter argument against my point, as it just points out a methodological problem. If we adopt your argument (b), then your definition of complexity/difficulty suddenly becomes dependent on the type of linear readout that you use. This is why I suggested to find the linear classifier on the test set as well. Assuming it's a convex problem (which is likely is in most of your cases) being unable to find a separable plane would imply that there is none and you can now adopt (A). However, if there is one for each layer because you have too few datapoints, then you cannot uphold (A) because your measure for difficulty basically becomes flat. I also don't think that your reply is a good counter argument in this case, as your learning algorithm simply fails to find the right hyperplane (maybe it would be better with a different regularizer). In addition, your definition of "difficulty" depends on the actual linear learning algorithm that you used (because there is a separating hyperplane that generalizes well; your algorithm just doesn't find it from the training data). I don't see how this connects to "difficulty" or "complexity" because not having enough data or using the wrong linear classifier shouldn't affect how complex or difficult a task is.

Now you could adopt another interpretation (B): Maybe a representation that is characteristic for a particular task should make that task easy to learn and generalize well. To give you an example for what I mean: Consider a finite degree polynomial on a bounded interval in the real line. You could try to learn that polynomial from a basis of monomials or from a basis of sinusoids. Both should work to arbitrary precision. However, since the original function is a polynomial its basis coefficients are sparse in the monomial basis but not in the Fourier basis. That's why it should be easier (more data efficient) to learn in the monomial basis than in the Fourier basis. In addition, it should extrapolate better. You seem to adopt a similar argument in your paper and response. In this case it's about how well the learned linear function generalizes and how data efficient it can be learned. However, in my opinion this does not really measure difficulty or complexity, but rather how characteristic a particular feature set is for that task (i.e. monomials are characteristic for polynomials but sinusoidals are not, in my example).

Since a lot of your conclusions hinge on the "difficulty" interpretation I ask you to clarify your line of reasoning in this respect. Again, from figure 1A you seem to equate difficulty with linear separability and we agree that you cannot detect this given the limited number of datapoints. Thus, the crucial question is: What do you exactly mean by difficulty and how is that connected to generalization of one particular linear learning algorithm?

Minor comments

- The quality of you plots is really low. I presume it's because PLoS, for some reason, wants/produces tiffs out of what was once a beautiful vector graphic. If that's not the case, please check you graphics. There are a lot of compression artifacts in them.

**Have all data underlying the figures and results presented in the manuscript been provided?**

Reviewer #1: Yes

Reviewer #2: **No: **The authors have provided a link for the computational software, but the linked page does not have the data and relevant software.

Reviewer #3: Yes

PLOS authors have the option to publish the peer review history of their article (what does this mean?). If published, this will include your full peer review and any attached files.

Reviewer #1: No

Reviewer #2: No

Reviewer #3: No
---

## [Editor Report · Decision Letter 2]

17 Jan 2021

Dear Dr. Vinken,

We are pleased to inform you that your manuscript 'Using deep neural networks to evaluate object vision tasks in rats' has been provisionally accepted for publication in PLOS Computational Biology.

Best regards,

Ronald van den Berg

Associate Editor

PLOS Computational Biology

Wolfgang Einhäuser

Deputy Editor

PLOS Computational Biology

---

## [Editor Report · Acceptance letter]

24 Feb 2021

PCOMPBIOL-D-20-01051R2 

Using deep neural networks to evaluate object vision tasks in rats

Dear Dr Vinken,

I am pleased to inform you that your manuscript has been formally accepted for publication in PLOS Computational Biology. Your manuscript is now with our production department and you will be notified of the publication date in due course.

With kind regards,

Alice Ellingham
